# Variation in the Ovine Glycogen Synthase Kinase 3 Beta-Interaction Protein Gene (*GSKIP*) Affects Carcass and Growth Traits in Romney Sheep

**DOI:** 10.3390/ani11092690

**Published:** 2021-09-14

**Authors:** Fangfang Zhao, Huitong Zhou, Shaobin Li, Qingming An, Qian Fang, Yuzhu Luo, Jon G. H. Hickford

**Affiliations:** 1Gansu Key Laboratory of Herbivorous Animal Biotechnology, Faculty of Animal Science and Technology, Gansu Agricultural University, Lanzhou 730070, China; zhaofangfang@gsau.edu.cn (F.Z.); Zhou@lincoln.ac.nz (H.Z.); lisb@gsau.edu.cn (S.L.); 2International Science and Technology Cooperation Base of Meat Sheep and Meat Cattle Genetic Improvement in Northwest of China, Gansu Agricultural University, Lanzhou 730070, China; Freeman.Fang@lincoln.ac.nz; 3Gene-Marker Laboratory, Faculty of Agricultural and Life Sciences, Lincoln University, Lincoln 7647, New Zealand; 4Faculty of Wujiang, Tongren University, Tongren 554300, China; anqingming2009@163.com

**Keywords:** NZ Romney sheep, glycogen synthase kinase 3 beta-interacting protein gene (*GSKIP*), nucleotide substitution, ovine carcass traits, ovine growth

## Abstract

**Simple Summary:**

The glycogen synthase kinase 3 beta (GSK3β)-interacting protein plays a role in regulating glycogen metabolism, protein synthesis, the cell cycle, and in regulating gene expression. To date, physiological function research into the GSK3β-interacting protein has been focused on cell lines, gene ‘knockout’ models, and over-expression studies, and to our knowledge, there have been no reports on how variation in the GSK3β-interacting protein gene (*GSKIP*) may affect phenotypic traits. In this study, PCR-SSCP methods were used to screen for variation in exon 1 and exon 2 of *GSKIP* in 840 New Zealand (NZ) Romney sheep. Two variant sequences were identified in exon 1 and this variation in *GSKIP* was associated with variation in lamb birth weight, hot carcass weight, and fat depth at the 12th rib.

**Abstract:**

The glycogen synthase kinase 3 beta (GSK3β)-interacting protein (encoded by the gene *GSKIP*) is a small A-kinase anchoring protein, which complexes with GSK3βand protein kinase A (PKA) and acts synergistically with cAMP/PKA signaling to inhibit GSK3β activity. The protein plays a role in regulating glycogen metabolism, protein synthesis, the cell cycle, and in regulating gene expression. In this study, PCR-single strand conformation polymorphism (PCR-SSCP) analyses were used to screen for variation in exon 1 and exon 2 of *GSKIP* in 840 New Zealand (NZ) Romney sheep. Two SSCP banding patterns representing two different nucleotide variants (*A* and *B*) were detected in an exon 1 region, whereas in an exon 2 region only one pattern was detected. Variants *A* and *B* of exon 1 had one non-synonymous nucleotide difference c.37A/G (p.Met13Val). The birthweight of sheep of genotype *AA* (5.9 ± 0.06 kg) was different (*p* = 0.023) to sheep of genotype *AB* (5.7 ± 0.06 kg) and *BB* (5.7 ± 0.06 kg). The hot carcass weight (HCW) of sheep of genotype *AA* (17.2 ± 0.22 kg) was different (*p* = 0.012) to sheep of genotype *AB* (17.6 ± 0.22 kg) and *BB* (18.0 ± 0.29 kg), and the fat depth at the 12th rib (V-GR) of sheep of genotype *AA* (7.7 ± 0.31 mm) was different (*p* = 0.016) to sheep of genotype *AB* (8.3 ± 0.30 mm) and *BB* (8.5 ± 0.39 mm). The results suggest that the c.37A/G substitution in ovine *GSKIP* may affect sheep growth and carcass traits.

## 1. Introduction

The New Zealand Romney (NZ Romney) is the most common sheep breed in New Zealand and underpins the production of lambs for the slaughter trade. In this production system, farmers are paid based on carcass weight and carcass grading for muscle and fat traits, hence the genetics of these traits is of interest to NZ Romney sheep breeders.

Glycogen synthase kinase 3 beta (GSK3β), a multifunctional protein kinase, is involved in glycogen metabolism [1], protein synthesis [2], the cell cycle [3], and gene expression [4], where it is a key regulator of insulin, growth factor, and Wingless and Int-1 (Wnt) signal transduction. Loss of regulation of GSK3β levels is associated with human diseases including cancer, Alzheimer’s disease, diabetes, and bipolar disorder [5,6].

A binding protein regulates the activity of GSK3β, and its inactivity promotes Wnt signaling and animal muscle and adipose tissue growth and development [7,8,9]. This binding protein, called the GSK3β-interacting protein, was originally recognized by its ability to inhibit the biological activity of GSK3β in the Wnt signaling pathway [10], but it is now known as an A-kinase anchoring protein (AKAP) family member and a cytosolic scaffolding protein that binds protein kinase A (PKA) and GSK3β [11].

The GSK3β-interacting protein is active at the start of the protein kinase B (PKB/AKT)/GSK3 and Wnt/beta-catenin signaling pathways. It facilitates control of the β-catenin through stabilizing phosphorylation by PKA at Ser-675, and its association with GSK3β helps to control of the destabilizing phosphorylation of β-catenin. The effects of GSK3β-interacting protein on β-catenin can be explained by its scavenger function: it recruits kinases from the destruction complex and avoids interacting with β-catenin forming complex [11].

Some researchers suggest that the GSK3β-interacting protein participates in neuron development [12], mitochondrial elongation [13], perinatal growth and development [14], and myeloid neoplasms [15]. To date, physiological function research into the GSK3β-interacting protein has been focused on cell lines, gene ‘knockout’ models, and over-expression studies, and to our knowledge, there have been no reports on how variation in the GSK3β-interacting protein gene (*GSKIP*) may affect phenotypic traits, including key growth and meat production traits in the livestock industries.

Chou et al. (2006) reported that human *GSKIP* has three exons and that the first exon is non-coding [16] and a three exon ovine *GSKIP* sequence is reported for the Sheep (Oar_rambouillet_v1.0) construct on ovine chromosome 18 between nucleotides 61,754,294 and 61,773,116.

In this study, PCR single strand conformation polymorphism (SSCP) analysis was used to screen variation in two regions of ovine *GSKIP* and ascertain its association with carcass and growth traits in New Zealand (NZ) Romney sheep. The first region encompasses exon 1 and contains sequences that potentially affect gene expression, and the second region encompasses exon 2, a region that likely spans the PKA-binding domain in sheep [11].

## 2. Materials and Methods

### 2.1. Sheep Studied and Data Gleaned

Eight hundred and thirty one male (n = 497) and female (n = 334) NZ Romney lambs were investigated, and these animals and their production data were obtained from a previous study based on a NZ Romney breeders’ progeny test [17]. The lambs were the progeny of 19 unrelated sire lines and raised on the same farm. Each ram had been randomly single-sire mated to 40–60 NZ Romney ewes that were ranged in age from four to seven years. All the sheep was ear-marked with a unique ID number within twelve hours after birth, and birth rank (number of full sib), rearing rank, the gender, and birth weight were recorded for every sheep. Blood samples from individual lambs were dripped onto TFN paper (Munktell Filter AB, Stockholm, Sweden) by nicking the ears of sheep.

The approach of blood collection of this study complied with Animal Identification of the Animal Welfare (Sheep and Beef Cattle) Code of Welfare 2010 Section 7.5, which is a welfare issued code under the Animal Welfare Act 1999 by New Zealand Government. In contemporary NZ farming, both blood and ear tissue samples are regularly collected for the purpose of animal breeding, and this can include the use of commercially available single-gene diagnostic tests for both disease and performance traits, and the use of genome wide screening approaches including SNP chip typing and whole genome sequencing. Accordingly, ethics approval was not directly required for this research.

All these sheep were weaned, weighed, and separated based on their gender at approximately 90 days of age. The growth rate to weaning of the lambs, meaning the average daily weight gain before weaning, was calculated using birth weight, weaning weight, and weaning age. As most of the female lambs were kept as ewe replacements for the base flock, the draft weight and carcass data were only available for most of the male lambs (n = 456; some were kept for stud/breeding purposes), and cull ewe lambs (n = 81; most culled for wool faults). Of these, lambs weighing 37 kg or more were drafted for slaughter. The remaining lambs were next weighed at 16 weeks of age and those that had then reached 37 kg were slaughtered. Finally, at 20 weeks of age, all remaining lambs were slaughtered. Weight at drafting was recorded for each lamb, and its post-weaning growth rate was calculated as the difference between draft weight and weaning weight, divided by age in days (expressed in grams/day). Hot carcass weights (HCW) were directly collected on the production line. The HCW is the weight in kilograms of the carcass without the head, pelt and internal organs. VIAscan^®^ Sheep Carcase System (VIAScan^®^; Sastek, Brisbane, Australia) was used to estimate carcass traits as described by Hopkins et al. [18], including subcutaneous fat depth near the 12th rib (V-GR), lean meat yield (expressed as a percentage of HCW) in the shoulder (shoulder yield), loin (loin yield), and leg (leg yield), and the total yield (the sum of the shoulder, loin, and leg yields). As most of the female lambs were retained as backup ewes to expand commercial base flock, the carcass traits were only gathered from the male lambs, and only a few female lambs.

### 2.2. The PCR Amplification of Ovine GSKIP

Genomic DNA on the TFN paper was purified for PCR analysis using a method described by Zhou et al. [19]. Two sets of PCR primers, 5′-TATAAGCCTCTTTCATACCC-3′ and 5′-CTAAAGGGAAAAGGG ACTTAC-3′ (ovine genome sequence NC_040269.1, 61768512-61768531 and 61768805-61768823, respectively), were used to amplify a 412-bp fragment containing exon 1 of ovine *GSKIP*. Another two primers, 5′-TTTGCTTTCTGTACCAAGATG-3′and 5′-CTATGACCGTATGCAGCAG-3′ (ovine genome sequence NC_040269.1, 61772615-61772633 and 61772890-61772908, respectively), were used to amplify a 294-bp fragment covering exon 2 of ovine *GSKIP*. The primers were synthesized by Integrated DNA Technologies (Coralville, IA, USA).

The PCR amplifications were conducted in a 15-µL reaction containing 0.5 U Taq DNA polymerase (Qiagen, Hilden, Germany), 0.25 µM of primer, 2.5 mM Mg^2+^, 150 µM dNTPs (Bioline, London, UK), 1× the reaction buffer, and a 1.2-mm punch of the TFN paper that containing purified genomic DNA. The PCR reaction was conducted as follows: 94 °C for 3 min, followed by 35 cycles of 94 °C for 30 s, 60 °C for 30 s, and 72 °C for 30 s, with a final elongation of at 72 °C for 5 min. S1000 thermal cyclers system (Bio-Rad, Hercules, CA, USA) was used to PCR amplification.

The amplicons productions were visualized after electrophoresing in agarose gels made by 1% agarose, 1× TBE buffer, and 200 ng/mL of ethidium bromide.

### 2.3. Screening for Variation in Ovine GSKIP

SSCP analysis was then used for screening sequence variation using the PCR amplicons. A 0.7-μL PCR amplicon and 7 μL of loading dye (0.025% xylene-cyanol, 98% formamide, 0.025% bromophenol blue, 10 mM EDTA) were mixed fully. First denaturation 5 min at 95 °C, then immediately ice bath for 2 min and then loaded the samples on 14% acrylamide: bisacrylamide (37.5:1) (Bio-Rad, CA, USA) gels. Electrophoresis for the exon 1 amplicons was performed using Protean II xi cells in 0.5× TBE buffer at 22 °C and 250 V for 15 h. For the exon 2 amplicons, electrophoresis was undertaken at temperatures varying from 4 °C to 30 °C in an attempt to find genetic variation. Gels were silver stained using the method described by Byun et al. [20].

### 2.4. Sequencing and Analysis of Ovine GSKIP Variants

Sequencing of ovine *GSKIP* variants that were detected as homozygous by SSCP analysis were conducted by Lincoln University Sequencing Facility, NZ. DNAMAN (version: 8.0.8, 789, Lynnon BioSoft, Vaudreuil, MN, Canada) was used for sequence assembly, alignments, and translations analysis.

### 2.5. Statistical Analyses

General Linear Mixed Models (GLMMs) were used to analyze the effect of the *GSKIP* genotype on growth and carcass traits using Minitab (version 17, Minitab Inc., State College, PA, USA). Tukey test with Bonferroni corrections was performed for multiple pairwise comparisons between *GSKIP* genotypes. Sire was set as a random explanatory factor for all GLMMs analysis. For the birth weight GLMMs, birth rank and gender were fitted as fixed explanatory factors, but for the growth to weaning and weaning weight GLMMs, rearing rank, and gender were treated as fixed explanatory factors. Weaning age was also fitted in weaning weight GLMMs as a co-variate. For all the carcass traits, draft age and birth weight were fitted as covariates, while gender was set as a fixed factor into the models. Only main effects were analyzed, and *p* < 0.05 was a significant level for association analysis.

## 3. Results

### 3.1. Ovine GSKIP Variation

In the ovine *GSKIP* exon 1 amplification region two PCR-SSCP banding patterns were observed, with the two basic bands along or combined with each other for each sample (Figure 1a). DNA sequencing showed that the two band patterns represented two different nucleotide sequences (named *A* and *B*). A single nucleotide polymorphism (SNP) was identified in the two sequences, and this was named c.37A/G. Sequence *A* was c.37A, sequence *B* was c.37G, and these sequences were submitted to GenBank and get two accession numbers MH144562 and MH144563 respectively. The SNP was non-synonymous and would result in an amino acid change p.Met13Val.

Only one SSCP pattern was detected under various electrophoresis temperatures for the *GSKIP* exon 2 amplicon (Figure 1b), suggesting there was no sequence variation in this region.

### 3.2. Frequencies of the Ovine GSKIP Variants in the NZ Romney Sheep

In the 840 NZ Romney lambs, three genotypes were observed for the exon 1 variation. These were *AA* (45.6%), *BB* (14.5%), and *AB* (33.9%), with the variant frequencies being 61.1% for *A* and 38.9% for *B*.

### 3.3. Effect of Ovine GSKIP Variants on Carcass and Growth Traits

Associations between the *AA*, *BB,* and *AB* genotypes of *GSKIP* and birth weight, hot carcass weight, and V-GR were revealed (Table 1). Sheep with genotype *BB* had lower mean birth weight (*p* = 0.023), but higher *HCW* (*p* = 0.012) and higher V-GR (*p* = 0.016) than those sheep of the *AB* or *AA* genotypes (Table 1).

## 4. Discussion

This is the first study reporting variation in ovine *GSKIP* and its association with growth and carcass traits. Only one SNP was detected in exon 1 and exon 2 of *GSKIP* in 840 NZ Romney sheep, with this suggesting that the nucleotide sequence of *GSKIP* is conserved. Highly conserved sequences are typically associated with proteins that underpin essential metabolic activities [17]. With an important role in the AKT/GSK3 and Wnt/beta-catenin signaling pathway, and the need to maintain stability in subsequent physiological functions, it therefore seems that a stable gene structure is needed, especially as exon 2 of *GSKIP* contains the start codon and encodes the PKA-binding domain [11].

Variation in *GSKIP* was found to be associated with three traits: birth weight, HCW, and V-GR. The birthweight of sheep of genotype *AA* (5.9 ± 0.06 kg) was different (*p* = 0.023) to sheep of genotype *AB* (5.7 ± 0.06 kg) and *BB* (5.7 ± 0.06 kg). The hot carcass weight (HCW) of sheep of genotype *AA* (17.2 ± 0.22 kg) was different (*p* = 0.012) to sheep of genotype *AB* (17.6 ± 0.22 kg) and *BB* (18.0 ± 0.29 kg), and the fat depth at the 12th rib (V-GR) of sheep of genotype *AA* (7.7 ± 0.31 mm) was different (*p* = 0.016) to sheep of genotype *AB* (8.3 ± 0.30 mm) and *BB* (8.5 ± 0.39 mm), noting these are all estimated marginal means derived from the GLMMs. In a previous study of the sheep studied here [17], lamb birth weight was revealed to have a weak positive correlation with V-GR (r = 0.223) and a moderate correlation with HCW (r = 0.362) [17], while HCW and V-GR had a strong positive correlation (r = 0.601). This might suggest that the effect of the *GSKIP* variation on one or more of these traits was detected because of the correlation of the traits, especially the highly correlated HCW and V-GR. Ascertaining whether the *GSKIP* effect on these traits was independent would require further analysis, not least ascertainment of how the GSK3β-interacting protein affects carcass fat deposition, and deposition in the various carcass depots including subcutaneous fat, intermuscular fat, and intramuscular fat.

There was no difference in birth weight, HCW, and V-GR between sheep of genotype *AB* and those of genotype *BB*, suggesting that *A* has a recessive effect, while *B* may have a dominant effect on these traits. This suggests that *A* is associated with an increased mean birth weight and a decrease in HCW and V-GR, while *B* is associated with a decrease in birth weight and an increase in HCW and V-GR. The consequences of this for lamb production will need further investigation. Birth weight can affect the survival of lambs: with low birth weight, lambs experience higher levels of postnatal mortality [21,22]. This suggests that selecting for *A* in sheep breeding would improve survival, but potentially at the cost of improved hot carcass weight, for which New Zealand Romney lamb producers are paid. Accordingly, the survival of lambs with variant *B* probably needs to be examined to ascertain whether they are compromised.

The c.37A/G SNP was non-synonymous, and if expressed it would lead to an amino acid change (p.Met13Val). This may cause a change to the GSK3β-interacting protein structure and function, although the amino acid change is outside of the putative PKA binding site [11]. This is not to say that it could not have an effect on protein structure and function. The AKT/GSK3 and Wnt/β-catenin signaling pathways regulate cell proliferation, differentiation, growth, and apoptosis as well as the expression of specific genes. It is therefore of importance for growth and the development of animals. As the protein is a key player in these pathways, sequence variation in its gene, especially non-synonymous nucleotide substitutions, may result in change of the expression and/or structure of protein and consequently affect growth and development. This is supported by the findings of Pegliasco et al. [15], whose study suggested that the protein plays a key role in early clonal hematopoiesis and human health.

## 5. Conclusions

This study used PCR-SSCP to detect variation in *GSKIP* and identified one SNP (c.37A/G) in NZ Romney sheep. This SNP was found to affect birth weight, HCW, and V-GR. Further investigation in more sheep from different breeds and age is required to confirm these results.

## Figures and Tables

**Figure 1 animals-11-02690-f001:**
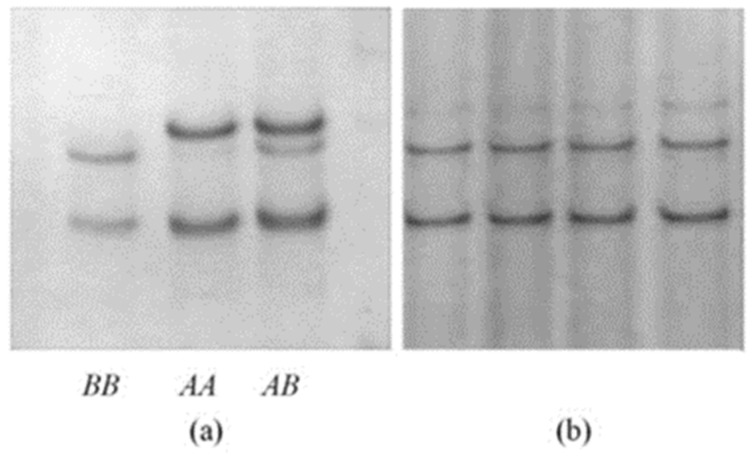
PCR–SSCP detection results of ovine *GSKIP*. “(**a**)” is SSCP results of exon 1 amplification region ovine *GSKIP*; “(**b**)” is SSCP results of exon 2 amplification region of ovine *GSKIP.*.

**Table 1 animals-11-02690-t001:** Association of ovine *GSKIP* genotypes with carcass and growth traits in sheep.

Trait ^Ⅰ^	Mean ± SE ^Ⅱ^			*p*
	*AA*	*AB*	*BB*	
	(n = 377)	(n = 332)	(n = 122)	
Birth weight (kg)	**5.9 ± 0.06 ^a^**	**5.7 ± 0.06 ^b^**	**5.7 ± 0.09 ^b^**	**0.023**
Weaning weight (kg)	33.6 ± 0.46	33.9 ± 0.46	34.2 ± 0.56	0.537
Growth rate to weaning (g/d)	319.9 ± 4.35	320.8 ± 4.42	323.9 ± 5.79	0.799
	(n = 236)	(n = 224)	(n = 77)	
HCW (kg)	**17.2 ± 0.22 ^b^**	**17.6 ± 0.22 ^a^**	**18.0 ± 0.29 ^a^**	**0.012**
V-GR (mm)	**7.7 ± 0.31 ^b^**	**8.3 ± 0.30 ^a^**	**8.5 ± 0.39 ^a^**	**0.016**
Shoulder yield (%)	17.3 ± 0.10	17.4 ± 0.10	17.3 ± 0.13	0.116
Loin yield (%)	14.9 ± 0.10	14.9 ± 0.10	14.9 ± 0.13	0.852
Leg yield (%)	22.2 ± 0.13	22.2 ± 0.13	22.0 ± 0.17	0.276
Total yield (%)	54.4 ± 0.27	54.5 ± 0.26	54.2 ± 0.34	0.546

^Ⅰ^ HCW is short for hot carcass weight; V-GR means VIAScan fat depth at the 12th rib. ^Ⅱ^ Predicted standard error of means derived from GLMMs. Different small letters in the same line indicate significant differences (*p* < 0.05), bold data mean *p* < 0.05.

## Data Availability

The two sequences of ovine *GSKIP* exon 1 alleles were submitted to GenBank and get two accession numbers MH144562 and MH144563 respectively.

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
