# Peer review of "Variation in the Ovine Glycogen Synthase Kinase 3 Beta-Interaction Protein Gene (GSKIP) Affects Carcass and Growth Traits in Romney Sheep"

_animals, 2021, doi:10.3390/ani11092690_

Round 1

Reviewer 1 Report

The authors presented a paper on variation in the ovine glycogen synthase kinase 3 beta-interaction protein gene (GSKIP) and its effects on carcass and growth traits in Romney sheep. Even if we are in the era of the genomic selection the approach of the candidate gene is also today interesting, especially for the potential transferability of the results. In my opinion the paper is worthy of publication, reporting for the first time a polymorphism in a gene that can affects productive traits; moreover, such polymorphism could be inclued in the selection schemes to improve both growth and carcass traits.

Some corrections are requested in order to make the paper more understandable and clear for the readers. More in details:

-lines 41-42: try to avoid words already included in the title.

-line 46: please supply a reference also for “gene expression”

-line 68: have you any information/reference about GSKIP in sheep?

-lines 91-93: it could be better to specify how many males and females. Were the animals reared in the same farm? Moreover, at what age were the animals slughtered? Were the animals slaughtered in the same slaughter house? Please specify all these aspects.

-lines 101-102: again, in a scientific manuscript it is important to specify the number of males and females.

-line 112: please move the sentence “Genomic DNA on the TFN paper was purified for PCR analysis using a method described by Zhou et al. [18].” at the start of the paragraph 2.2

Author Response

Thank you for you valuable comments. We have marked our changes in the manuscript in red text.

Comments and Suggestions for Authors The authors presented a paper on variation in the ovine glycogen synthase kinase 3 beta-interaction protein gene (GSKIP) and its effects on carcass and growth traits in Romney sheep. Even if we are in the era of the genomic selection the approach of the candidate gene is also today interesting, especially for the potential transferability of the results. In my opinion the paper is worthy of publication, reporting for the first time a polymorphism in a gene that can affects productive traits; moreover, such polymorphism could be inclued in the selection schemes to improve both growth and carcass traits. Some corrections are requested in order to make the paper more understandable and clear for the readers.

More in details: -lines 41-42: try to avoid words already included in the title.

AU: We have changed some of the keywords.

-line 46: please supply a reference also for “gene expression”

AU: A reference has been added.

-line 68: have you any information/reference about GSKIP in sheep?

AU: We have added an Ensembl reference to ovine GSKIP (A three exon ovine GSKIP sequence is reported for the Sheep (Oar_rambouillet_v1.0) construct on ovine chromosome 18 between nucleotides 61,754,294 and 61,773,116).

-lines 91-93: it could be better to specify how many males and females. Were the animals reared in the same farm? Moreover, at what age were the animals slughtered? Were the animals slaughtered in the same slaughter house? Please specify all these aspects.

AU: A lot more detail has been added.

-lines 101-102: again, in a scientific manuscript it is important to specify the number of males and females. -line 112: please move the sentence “Genomic DNA on the TFN paper was purified for PCR analysis using a method described by Zhou et al. [18].” at the start of the paragraph 2.2

AU: Sentence moved.

Reviewer 2 Report

Title: The title of the manuscript is clear and links well with the rest of the paper

Abstract: The abstract has failed to capture upfront the main hypothesis being tested. Experimental design should be based on hypothesis being tested, but in this study, there was no clearly defined hypothesis.

The author needs to report the actual numbers and the p-values in the abstract.

Introduction: The introduction is poorly written. It fails to clearly capture upfront the main purpose of the study that clearly links with the other sections in the manuscript. I find the introduction lacking direction and does not clearly teases out the problems with New Zealand Romney sheep that needs addressing urgently. Why is it important that we the readers need to know about “Variation in the ovine glycogen synthase kinase 3 beta-interaction protein gene (GSKIP) affects carcass and growth traits in Romney sheep?” This has not been clearly addressed in the introduction. I’m assuming that this study will be important in farming, genotyping and breeding of New Zealand Romney sheep? If so, this has to come out clearly in the introduction. The author was loquacious on GSK3β with no clear direction to where the introduction was heading, denying flow to the story.

Wnt and PKA should be explained at first mention.

Lines 66-67 “and to our knowledge, there have been no reports on how variation in the GSK3β-interacting protein gene (GSKIP) may affect phenotypic traits” the author should be careful here. This study is only looking at exons 1 & 2, and not the entire gene sequence. This statement should be rewritten to include only exons 1 & 2.

Lines 70-72 “In this study, PCR- single strand conformation polymorphism (SSCP) analysis was used to screen variation in exon 1 and exon 2 of ovine GSKIP, and ascertain its association with carcass and growth traits in New Zealand (NZ) Romney sheep”. The author should justify why they chose exons 1 & 2 while they had mentioned that “GSKIP has three exons and that the first exon is non-coding”. Why didn’t the author choose exon 3 instead of 1 knowing that 1 was non-coding?

Materials and Methods:  Lines 83-90 should be the first paragraph to this section. Why was 840 New Zealand Romney sheep required for this study? Why was 19 non-related sire lines used? And ewes of 4-7 years? Please explain why ethic approval was not directly required for this study. Ethic approval is very important for any animal study. Acronym TFN needs to be explained at first mention. At what age were the animals slaughtered and why?

Lines 112-115 should be the starting paragraph of section 2.2.  Sections 2.3 and 2.4 are good.

Statistical Analysis: This section is good, but the author needs to expand on how the raw data was scrutinised for error. Was parametric or non-parametric tests conducted? Why did you use General Linear Mixed Models?

Results: Good. I can’t see a correlation table?

Discussion: There is no flow of idea/s. The author is all over the place. No consistency in reporting. The author needs to organise their thoughts better to tease out and present the key findings in this study in a concise and cogent style.

Lines 188-191: “Variation in GSKIP was found to be associated with three traits, birth weight, HCW and V-GR. Birth weight had a weak positive correlation with V-GR (r = 0.223) in the sheep studied and a moderate correlation with HCW (r = 0.362) [16], while HCW and V-GR had a strong positive correlation (r = 0.601)” Where is this result? This statement should be reported in the Results section 3.

This study is novel, but the author has failed to clearly demonstrate the significant impact their current findings might have on the sheep industry, especially the Romney sheep. 

Conclusion: The author has fail to present the implication(s) of their study’s results to New Zealand sheep industry. How is their study going to fit/address the challenges present in New Zealand sheep industry?

Author Response

Title: The title of the manuscript is clear and links well with the rest of the paper

Abstract: The abstract has failed to capture upfront the main hypothesis being tested. Experimental design should be based on hypothesis being tested, but in this study, there was no clearly defined hypothesis.

The author needs to report the actual numbers and the p-values in the abstract.

AU: The marginal means, standard errors of those means and p-values for the significant results have been added to the abstract, while the summary has been further simplified.

Introduction: The introduction is poorly written. It fails to clearly capture upfront the main purpose of the study that clearly links with the other sections in the manuscript. I find the introduction lacking direction and does not clearly teases out the problems with New Zealand Romney sheep that needs addressing urgently. Why is it important that we the readers need to know about “Variation in the ovine glycogen synthase kinase 3 beta-interaction protein gene (GSKIP) affects carcass and growth traits in Romney sheep?” This has not been clearly addressed in the introduction. I’m assuming that this study will be important in farming, genotyping and breeding of New Zealand Romney sheep? If so, this has to come out clearly in the introduction. The author was loquacious on GSK3β with no clear direction to where the introduction was heading, denying flow to the story.

AU: The introduction has been modified to improve the context.

Wnt and PKA should be explained at first mention.

AU: Definitions have now been added.

Lines 66-67 “and to our knowledge, there have been no reports on how variation in the GSK3β-interacting protein gene (GSKIP) may affect phenotypic traits” the author should be careful here. This study is only looking at exons 1 & 2, and not the entire gene sequence. This statement should be rewritten to include only exons 1 & 2.

AU: This sentence has been changed and the region of GSKIP analysed is described six lines further on.

Lines 70-72 “In this study, PCR- single strand conformation polymorphism (SSCP) analysis was used to screen variation in exon 1 and exon 2 of ovine GSKIP, and ascertain its association with carcass and growth traits in New Zealand (NZ) Romney sheep”. The author should justify why they chose exons 1 & 2 while they had mentioned that “GSKIP has three exons and that the first exon is non-coding”. Why didn’t the author choose exon 3 instead of 1 knowing that 1 was non-coding?

AU: The two exons were chosen for different reasons. The first region encompassed exon 1 and contains sequences that potentially affect gene expression, and the second region encompassed exon 2, a region that likely spans the PKA-binding domain. Detail to this effect has been added to the introduction.

Materials and Methods:  Lines 83-90 should be the first paragraph to this section. Why was 840 New Zealand Romney sheep required for this study? Why was 19 non-related sire lines used? And ewes of 4-7 years? Please explain why ethic approval was not directly required for this study. Ethic approval is very important for any animal study. Acronym TFN needs to be explained at first mention. At what age were the animals slaughtered and why?

AU: Detail on the ethics approval question has been added. Formal approval for observational studies of this kind is not required in New Zealand as the Animal Welfare (Sheep and Beef Cattle) Code of Welfare 2010 Section 7.5, which is a welfare issued code under the Animal Welfare Act 1999 by New Zealand Government, covers the use of ear nicking for blood collection and identification purposes.

The numbers of lambs and rams reflects the stated fact that the DNA and phenotypic data had been collected for another study (see reference [17]). This was an observational study and the sheep were part of an industry-based progeny test run by NZ Romney sheep breeders.

TFN is the brand name of a Munktell product. We cannot ascertain the etymology of the brand name.

Lines 112-115 should be the starting paragraph of section 2.2.  Sections 2.3 and 2.4 are good.

AU: This section has been modified and rearranged.

Statistical Analysis: This section is good, but the author needs to expand on how the raw data was scrutinised for error. Was parametric or non-parametric tests conducted? Why did you use General Linear Mixed Models?

AU: The phenotypic data was not tested for normality of distribution and this is a weakness in the model. The benefit of using GLMMs is they are useful for the analysis of data with more than one source of random variability. For example growth and effects thereon was measured a number of times over a period of time.

Results: Good. I can’t see a correlation table?

AU: Pearson correlation coefficients for the various traits are presented in reference [17] in which the same group of sheep were analysed, albeit in the context of variation in another gene. 

Discussion: There is no flow of idea/s. The author is all over the place. No consistency in reporting. The author needs to organise their thoughts better to tease out and present the key findings in this study in a concise and cogent style.

AU: The discussion has been majorly revised.

Lines 188-191: “Variation in GSKIP was found to be associated with three traits, birth weight, HCW and V-GR. Birth weight had a weak positive correlation with V-GR (r = 0.223) in the sheep studied and a moderate correlation with HCW (r = 0.362) [16], while HCW and V-GR had a strong positive correlation (r = 0.601)” Where is this result? This statement should be reported in the Results section 3.

AU: The section has been re-written to clarify that these correlations have been reported previously in reference [17].

This study is novel, but the author has failed to clearly demonstrate the significant impact their current findings might have on the sheep industry, especially the Romney sheep. 

Conclusion: The author has fail to present the implication(s) of their study’s results to New Zealand sheep industry. How is their study going to fit/address the challenges present in New Zealand sheep industry?

AU: The importance of the research for NZ sheep industry has been added at the beginning of introduction, as follows: “The New Zealand Romney (NZ Romney) is the most common sheep breed in New Zealand and underpins the production of lambs for the slaughter trade. In this production system, farmers are paid based on carcass weight and carcass grading for muscle and fat traits, hence the genetics of these traits is of interest to NZ Romney sheep breeders.”

Round 2

Reviewer 2 Report

I'm satisfied with your response. Please try and find another reference instead of the work of Deak et al. [15] that was done in mice. This reference provides evidence for a key part of the discussion.

Author Response

The reference of Deak et al was cited many times in our manuscript in the part of GSKIP-PKA binding site and its importance for animal growth and development.   Now we made a distinction, (1) found a new reference(Pegliasco et al. Germline ATG2B/GSKIP-containing 14q32 duplication predisposes to early clonal hematopoiesis leading to myeloid neoplasms.Leukemia,2021) to prove thier importance of human healthy; (2) use [11] to confirm the GSKIP-PKA binding site.  Please see the attachment and check the modification.  
